# A Gut Instinct on Leukaemia: A New Mechanistic Hypothesis for Microbiota-Immune Crosstalk in Disease Progression and Relapse

**DOI:** 10.3390/microorganisms10040713

**Published:** 2022-03-25

**Authors:** Ilaria S. Pagani, Govinda Poudel, Hannah R. Wardill

**Affiliations:** 1Cancer Program, Precision Medicine Theme, South Australian Health & Medical Research Institute, Adelaide 5000, Australia; Govinda.Poudel@sahmri.com (G.P.); hannah.wardill@adelaide.edu.au (H.R.W.); 2Faculty of Health and Medical Sciences, School of Medicine, University of Adelaide, Adelaide 5000, Australia; 3Faculty of Health and Medical Sciences, School of Biomedicine, University of Adelaide, Adelaide 5000, Australia

**Keywords:** gut microbiota, dysbiosis, chronic myeloid leukaemia, acute lymphoblastic leukaemia, disease progression, cancer relapse, immune suppressors, gut metabolites

## Abstract

Despite significant advances in the treatment of Chronic Myeloid and Acute Lymphoblastic Leukaemia (CML and ALL, respectively), disease progression and relapse remain a major problem. Growing evidence indicates the loss of immune surveillance of residual leukaemic cells as one of the main contributors to disease recurrence and relapse. More recently, there was an appreciation for how the host’s gut microbiota predisposes to relapse given its potent immunomodulatory capacity. This is especially compelling in haematological malignancies where changes in the gut microbiota have been identified after treatment, persisting in some patients for years after the completion of treatment. In this hypothesis-generating review, we discuss the interaction between the gut microbiota and treatment responses, and its capacity to influence the risk of relapse in both CML and ALL We hypothesize that the gut microbiota contributes to the creation of an immunosuppressive microenvironment, which promotes tumour progression and relapse.

## 1. Introduction

Chronic Myeloid and Acute Lymphoblastic Leukaemia (CML and ALL) are clonal haematological malignancies affecting the blood and bone marrow. Despite enormous progress in treatment regimens, disease relapse is still a major problem. The treatment of CML with Tyrosine Kinase Inhibitors (TKIs), targeting the BCR-ABL oncoprotein, has been paradigm shifting; they converted CML from a fatal to a chronic disease for most patients [1]. However, this is only partially successful with many patients needing continued, life-long therapy that come with physical, psychological, and financial costs which impair quality of life [2,3]. About 20% of CML patients do not respond to frontline therapy, with approximately half of them dying due to the disease progressing toward its final stage called “blast crisis”, or to complications post-allogeneic haematopoietic stem cell transplantation (HSCT) [4]. The discovery in 2006 that patients with exceptional responses to TKI therapy, can cease their life-long treatment, achieving a long-term remission in the absence of therapy (“treatment-free remission”) revolutionised the management of CML [4,5,6,7]. Treatment-free remission is now the therapeutic goal for CML patients. However, only ~25% of CML patients achieve treatment-free remission and maintain it for years. About 50% of patients who attempt therapy cessation relapse, generally after 3–6 months, and need to restart their TKI [5,6].

ALL is the most frequent cause of cancer-related death in childhood [8,9]. Whilst the understanding of the genomic alterations in ALL has allowed the development of new therapies, disease recurrence is observed in approximately 15–20% of paediatric and in 40–75% of adult patients [10,11,12,13]. Additional therapies are available for patients who failed chemotherapy or had suboptimal responses. They include HSCT, and immunotherapy targeting B-cells specific antigens such as CD19, CD20, and CD22 [14,15]. Monoclonal antibodies and their derivatives, and cellular immunotherapy using patient-derived T cells, modified with chimeric antigen receptors (CAR-T cells) are available for the treatment of relapsed and refractory leukaemia ALL [16]. Despite a successful initial response, the overall survival of patients treated with CART-T cells reaches only 13 months [17].

The gut microbiota, which comprises the metropolis of bacteria, fungi, viruses, archaea, and protozoa living in the gastrointestinal tract, affects human health, existing in a multi-layered symbiosis with the host. The host offers a vital habitat for these microorganisms, and in counterpart, the gut microbiota regulates the host metabolism, and contributes to shape the host’s immune and neuronal system [18,19,20,21]. While highly robust, the gut microbiota is extremely plastic and easily modified both beneficially and detrimentally. Environmental incursions such as use of antibiotics, changes in diet or geography, stress, medications, and co-morbidities all influence the composition of the gut microbiota, changing its metabolic profile and impacting on the host physiology. In the context of cancer therapy, chemotherapy, radiotherapy, and TKIs have all been reported to change the composition of the gut microbiota inducing a dysbiosis phenotype [22,23,24,25]. While the specific microbial phenotypes differ across different treatment modalities, they are generally characterised by a reduction in microbial diversity, a loss of beneficial bacteria and concomitant increase in pathological bacteria. These changes are associated with altered immune responses, emphasising the immunomodulatory capacity of the microbiota, which were already implicated in disease progression and relapse [19,20,21,23]. In this hypothesis-generating review we describe how residual leukaemic cells escape from the control of the immune system, and we focus on how the gut microbiota contributes to create an immunosuppressive microenvironment which is favourable to cancer progression and relapse in both CML and ALL.

## 2. Gut Microbiota and Regulation of the Host Immune System in Homeostasis

A healthy gut microbiota is mainly composed by the phyla of *Firmicutes, Bacteroidetes, Proteobacteria*, *Actinobacteria,*
*Fusobacteria*, and *Verrucomicrobi,* with the two phyla of *Firmicutes* and *Bacteroidetes* representing 90% of gut microbiota [26]. The gut microbiota exerts considerable immunomodulatory effects on the host, dictating the fine balance between response and tolerance. Importantly, this potent immunomodulation can be observed locally at the site of the intestinal mucosa, and systemically with changes in circulating cytokines linked with the gut microbiota [27]. For example, studies performed using germ-free mice (mice devoid of a microbiota) demonstrate that the absence of commensal microbes was associated with: (i) an altered gene expression profile of the intestinal epithelium, (ii) defects of lymphoid tissue architecture and immune function, (iii) a reduction in Immunoglobulin A (IgA)-plasma cells [28], and an imbalanced ratio of T-helper 17 (Th17) and Regulatory T (Treg) cells [29,30]. Germ-free mice are also characterised by smaller hematopoietic stem and progenitor cell populations and by a reduced number of myeloid cells compared with their specific non-germ-free counterparts [31].

The interactions between the microbiota and the host’s immune system are strictly compartmentalised at the level of the intestinal epithelium (Figure 1A). The epithelium acts as a barrier between the microbial content of the intestinal lumen and the underlying mucosal immune system, and it is also referred to as the “intestinal barrier”, or epithelial mucus layer. It consists of glycoproteins, mucins, immunoglobulins, and butyrate. The microbiota contributes to maintain this barrier [32]. To prevent the intestinal barrier function from being breached, *Lactobacillus* (*Firmicutes* phylum) forms a biofilm covering the enterocytes, and in this way isolates the pathogen-associated receptors here located [33]. The host innate immune system cells interact with the microbial metabolites at the level of the intestinal barrier, at the host-microbiome interface, “sensing” the microorganisms. They translate the microbial signals into host physiological responses and, at the same time, regulate the microbial ecology [19]. The bidirectional dialogue between the gut microbiota and the host normally occurs through the production and regulation of metabolites, including essential amino acids, vitamins, hormones, and short chain fatty acids (SCFAs), derived from the fermentation of non-digestible substrates, including fibres [34]. SCFAs are saturated organic aliphatic acids, including acetate, propionate, and butyrate. They are not only involved in the maintenance of the intestinal epithelial barrier, but also directly modulate the host immune function and inflammatory responses [34]. It was proposed that SCFAs have anti-inflammatory effects through direct activation of free fatty acid receptors (G-proteins coupled receptors, GPRs) [35]. SCFAs suppress the secretion of pro-inflammatory cytokines, such as Interleukin (IL)-1β, IL-6 and IL-8, by inhibiting the Nuclear Factor-kB (NF-kB) signalling pathway, and regulate several intracellular signalling pathways including Mitogen-Activated Protein Kinases (MAPKs: ERK, c-Jun N-terminal kinase (JNK) and p38 MAPK)) [36]. Additionally, pattern recognition receptors, including Toll-Like Receptors (TLRs), have an important role in sensing the microbial signals. Initially described for their key role in infections, they are also produced by the microbiota during the host colonization. TLRs initiate the host innate immune response in response to pathogens, regulate the microbial homeostasis, and maintain tissue integrity (Figure 1A) [37].

The gut microbiota further controls host immunity through epigenetic changes which can ultimately alter the Th17/Treg balance [38,39]. SCFAs can inhibit histone deacetylase activity and promote histone acetylation at the promoter of Signal Transducer and Activator of Transcription 3 (*STAT3)*, *NF-kB* and Forkhead box protein P3 (*FOXP3)*, thereby enhancing the number of Treg and decreasing the number of Th17 cells [36,40,41]. Of note, uncontrolled Th17 cells can induce severe autoimmune diseases, including inflammatory bowel disease, rheumatoid arthritis, multiple sclerosis, psoriasis [42]. On the other hand, Treg are immunosuppressors, and therefore maintain homeostasis and are involved in self-tolerance preventing autoimmune disease [43]. They produce the anti-inflammatory cytokines IL-10 and Transforming Growth Factor beta (TGF-β) and inhibit immune responses. Importantly, dysregulated Treg are involved in in cancer progression, infiltrating the tumour and inhibiting antitumour immunity [44,45]. This intense ability of the microbiota to control the fine balance that exists between tolerance and response is of particular relevance in CML and ALL relapse, as dysregulated immune surveillance is known to impair recognition and eradication of residual leukemic cells that evaded initial treatment [46,47,48,49,50].

## 3. Immune-Mediated Mechanisms That Drive Relapse in CML and ALL

Dysregulation of the immune system was reported during the progression of both CML and ALL, with an immune suppressive phenotype associated with relapse. Recent data indicate that failure of immunological surveillance of the residual leukemic cells was associated with relapse after TKI therapy cessation in CML. Several studies have reported that a lower absolute number of immune effectors (natural killer, NK cells), and higher number of immune suppressors (FOXP3+ Treg+ cells and monocytic myeloid-derived suppressor cells, mo-MDSCs) measured at the time of TKI therapy discontinuation was associated with subsequent relapse (Figure 2) [51,52,53,54,55]. An analysis of quiescent CML stem cells, persisting despite TKI therapy, identified markers of inflammation, including overexpression of IL-6, TGF-β, and Tumour Necrosis Factor alpha (TNF-α) [46,47,56,57]. Following therapy cessation, without the selective pressure induced by TKIs and in the presence of an immune compromised immune system, these residual leukaemic stem cells can exit quiescence and drive a relapse (Figure 2).

In a similar mechanism to CML, ALL recurrence is thought to be caused by the expansion of treatment resistant cells which evade initial treatment, later progressing to relapse. An important factor in the aetiology of ALL is the way in which leukemic cells interact with the microenvironment and reshape the bone marrow niche to create an immunosuppressive microenvironment and subsequent leukaemia progression [10]. Growing data indicate an impairment in function and number of effector immune cells, including NK cells, T cells, and macrophages, and an increase in immunosuppressors, such as Treg and granulocytic-myeloid derived suppressor cells (G-MDSCs). Tregs secrete inhibitory cytokines, suppressing the cytotoxic activity of T cells and reducing macrophage phagocytosis [10]. G-MDSCs produce Reactive Oxygen Species (ROS) and inhibits NK cells activity [58]. In the bone marrow niche mesenchymal stromal cells additionally sustain the growth of the leukaemic clones, through secretion of chemokines, NF-kB and metabolites such as asparagine, which suppresses the cytotoxicity of the L—asparaginase [59]. High-risk ALL subtypes also have an increased number of immune-checkpoint molecules, and correlate with decreased relapse-free survival.

These findings underscore the critical role that the immune system plays in regulating CML and ALL development and risk of relapse. With the growing appreciation for the bidirectional communication between the immune system and gut microbiota, this also prompted interest in how dysbiosis may contribute to failed immune surveillance and thus relapse. In addition, evidence also exists highlighting the carcinogenic effects of the gut microbiota (described in detail in the following section). When considering the dysbiotic changes in the gut microbiota in CML/ALL patients, it is therefore reasonable to suggest that persistent changes in the gut microbiota contribute to disease progression and relapse.

## 4. Gut Microbiota and Carcinogenesis

To live in harmony with the metropolis of microorganisms that inhabit our guts, the host immune system works to ensure commensal bacteria are tolerated, and harmful species are eliminated. People with chronic diseases, such as inflammatory bowel disease, type 2 diabetes, obesity, and cardiovascular disease, have been widely identified as having an altered gut microbiota [19,60]. These patients often lack bacteria populations required to activate the immune cells that in turn block the response against harmless bacteria. Cancer itself is also associated with differences in the composition of the gut microbiota, with unique microbial phenotypes reported in people with colorectal cancer [38] breast, [61,62] pancreas, [63] and hepatocellular carcinoma [64]. In the context of haematological malignancies, differences in the composition of the microbiota have been identified in CML and ALL patients, hypothesised to dictate the risk of disease and treatment efficacy [23,65,66,67,68,69]. Yu D et al., reported changes in the microbiota of 17 CML patients compared to normal health control, with higher relative abundances of genus *Streptococcus* and *Ruminococcus torques* group, and decrease in the relative abundance of genus *Bacteroides, Ruminococcaceae*, *Megamonas*, *Lachnospiraceae*, and *Prevotella* [68]. An interesting study performed by Rajagopala SV et al. showed a reduction in microbial diversity in paediatric and adolescent patients with ALL at the time of disease diagnosis in comparison with their healthy siblings [70]. Butyrate-producing *Lachnospiraceae* (which comprises the *Clostridium* XIVa and IV and *Roseburia*) were greatly reduced, but the *Bacteroidaceae* were increased, with authors suggesting these changes were involved in disease development [70]. This was also investigated experimentally, with an intact gut microbiota hypothesised to protect genetically predisposed mice from developing ALL [71]. Mice carrying *Pairing box 5* (*Pax5)* heterozygosity and the *ETV6-RUNX1* fusion, which predispose to ALL, did not develop leukaemia in a specific pathogen-free environment, in comparison with mice raised in a conventional facility where exposure to environmental pathogens is high. Additionally, haematopoietic cells carrying a genetic predisposition shaped a “genotype-specific” gut microbiota, affected B cells maturation, and altered the plasma metabolome. Interestingly, B-ALL development was also triggered by transient depletion of the microbiota through antibiotics. This indicates that is the alteration of the bacterial composition, in the presence of genetic predispositions, to trigger ALL, even in the absence of an infectious environment [71].

Although it remains debated as to whether the changes in the gut composition are causally involved in cancer, or simply a consequence of a disease process, it was identified that certain microorganisms promote carcinogenesis directly, by secreting metabolites that damage the host’s DNA [72]. For example, *Helicobacter hepaticus*, *Enterococcus faecalis,* and *Bacteroides fragilis* trigger the release of nitric oxide from immune cells, [73] ROS [74] and enterotoxin which activates the oncogenic driver *c-MYC*, [75] respectively. Because of this inflammatory status and enhanced microbial translocation, T cells become activated and differentiate into an inflammatory phenotype, resulting in Th17/Treg cells imbalance, with an increased number of immune suppressors Tregs (Figure 1B) [30]. This underscores the microbiota’s capacity to influence carcinogenesis through modulation of the immune system, and thus the potential to influence disease progression and relapse.

## 5. Microbial Dysbiosis in Cancer Progression and Relapse

### 5.1. Changes in the Gut Microbiota during Treatment and Association with Treatment Toxicity

Whilst cancer itself is associated with changes in the gut microbiota, the treatment of cancer is undoubtedly more damaging to the microbiota. Chemotherapy, radiotherapy, TKIs, and immunotherapy have all been shown to detrimentally impact the composition of the gut microbiota, with changes in its composition identified years after diagnosis [76,77,78,79,80,81]. These changes are amplified by high rates of antibiotic use, disease-associated stress, and changes in dietary habits [22,82,83]. These changes in the microbiota caused by cancer treatments are well-described, and while there are differences in the exact microbial phenotypes induced by different anti-cancer drugs, they are unified by common traits including a loss of overall diversity and a decrease in commensal microbes. These changes impair colonisation resistance, permitting the subsequent expansion of opportunity enteric pathogens largely belonging to the Proteobacteria phylum. Recent work demonstrated that butyrate-producing bacteria, such as *Roseburia*, *Coprococcus,* and *Faecalibacterium* are reduced in patients undergoing radiotherapy or cytotoxic chemotherapy [34]. In parallel, lipopolysaccharide-producing bacteria are increased, interacting with TLRs to activate the *NF-kB* signalling pathway and inducing intense inflammatory reactions in the gut (“mucositis”) (Figure 1B) [23,84]. This inflammatory injury of the intestinal mucosa disrupts tight junctions and increases intercellular spaces compromising the epithelial barrier [34,85]. This permits the translocation of pathogenetic bacteria and exacerbate proinflammatory immune responses and associated oxidative stress [86]. These changes cause several treatment side effects including infection, cachexia, graft versus host disease and even cognitive impairment, and are generally associated with poorer treatment outcomes [87,88,89,90]. Lipopolysaccharide is a potent inducer of inflammation, as testified by increased levels of ROS, TLR4, inflammatory cytokines, including IL-10, IL-6, TNF-α, IL-1β, and activation of the NF-kB pathway in K562 CML cell lines [91]. Interestingly, *Takizawa H* et al., showed that lipopolysaccharide, same as well the systemic infection with *Salmonella typhimurium*, was able to activate the proliferation of dormant haematopoietic stem cells through activation of TLR4-p38 MAPK pathway, and to increase death in mice following treatment with 5-fluorouracil (a cytotoxic chemotherapeutic agent used in the treatment of colorectal cancer, targeting proliferating cells). This highlights the ability of haematopoietic stem cells to sense pathogens and to regulate their proliferation and inflammatory responses accordingly [92].

Changes in the microbiota after treatment were identified after CML/ALL treatment, and in some cases, reported to persist for years after diagnosis [76,77,78,79,80,81]. During the first 6 weeks of induction therapy for ALL, the abundance of *Streptococcaceae* and *Enterococcaceae* increases while abundance of *Ruminococcacea* decreases [77,93,94]. Overall bacterial load decreases with consolidation and maintenance therapy phase, the abundance of *Enterococcaceae*, *Clostridiaceae,* and *Streptococcaceae* families increase while *Lachnospiraceae* and *Bifidobacteriaceae* families decrease [94]. The domination of the luminal environment with *Enterococcaceae* and *Streptococcaceae* was associated with increased risk of subsequent febrile neutropenia and diarrhoea, or solely diarrhoea, respectively [95]. The reconstitution of gut microbiota diversity often occurs after chemotherapy cessation and differs to the recovery of the microbiota’s taxonomic structure, which, instead, may remain altered long term [95]. Chua LL et al. showed that the stool samples of ALL survivors was enriched of *Actinobacteria* and depleted of *Faecalibacterium*, which produces butyrate and anti-inflammatory compounds [78]. This dysbiosis led to an increase in plasma levels of C-reactive protein, IL-6, and HLA-DR+CD4+ effector memory T cells and HLA-DR+CD8+ Treg cells subset [78]. There is some speculation that these chronic microbial changes may drive late effects in leukaemia survivors, including cardiovascular, metabolic, and neurocognitive diseases, [96] which have clear immunological drivers. As such, restoring the microbiota after treatment may offer a new strategy to mitigate the chronic morbidity caused by cancer therapy.

In instances where induction therapy is ineffective, people with ALL/CML may be offered HSCT [10,97]. This treatment involves the use of exceptionally high-dose chemotherapy, which due to its highly immunosuppressive nature, is often coupled with intensive antibiotic prophylaxis and empirical use. Oral intake is often reduced after HSCT, and total parenteral nutrition provided. As such, the gut microbiota faces numerous insults in HSCT recipients and undergoes profound, and often detrimental changes broadly characterised by a loss in commensal microorganisms and luminal domination of enteric pathogens. These changes were recently described to drive acute infectious complications, e.g., bacteraemia and fever, as well as chronic consequences including graft versus host disease [98,99]. This new knowledge prompted enthusiastic investigation of microbiota-targeted interventions, including faecal microbiota transplant, to prevent graft versus host disease [100].

### 5.2. Microbial Dysbiosis and Relapse

The events leading to the relapse of ALL and CML, especially in the absence of treatment, remain poorly understood. It is evident that leukaemic cells that persist undetected during treatment are likely to be the source of relapse, able to evade various treatments and the host’s immune system. While these cells have certainly developed ways to avoid immune detection, it is increasingly understood that the host’s immune system is also dysfunctional, and it is this insight that prompted considerations for how the microbiota may also contribute to relapse.

In light of the immune system’s important role in tumour surveillance, and the gut microbiota’s well-described impact on immune function, it is plausible to suggest that the microbiota is also involved in relapse [23,69,101]. This is especially compelling given the numerous insults the microbiota faces during ALL/CML treatments which leave the patient with chronic changes in its composition. The idea that the microbiota may influence cancer relapse is gaining theoretical and experimental traction. Several reports indicated the role of microbial metabolites in cancer progression and relapse. Lipopolysaccharide, for example, increases metastases in the lungs of mice, following activation of the NF-kB pathway, [102] and to increase malignancy in prostate cancer epithelial cells [103]. On the other end, the microbial metabolite sodium butyrate, a histone deacetylase inhibitor, increased the response of CD8+ T cells through IL-12 signalling, improving the efficacy of chemotherapy [104]. Propionate, another SCFA, was effective in reducing proliferation in BCR-ABL-expressing Baf3 cells through the activation of the free fatty acid receptor 2 [105]. This highlights the potential of microbial-derived metabolites in the treatment of cancer, described more in detail in the conclusive remarks section.

Furthermore, it was reported that antibiotics increase relapse, presumably through their impact on the microbiota [106,107]. For example, the use of azithromycin during the early phase of allogeneic-HSCT, to control bronchiolitis obliterans syndrome, is associated with increased incidence of haematological relapse and death [106]. In fact, the ALLOZITHRO clinical trial studied the effects of long-term therapy with azithromycin delivered from the early phases of the allogeneic-HSCT, but the study was prematurely ceased after 13 months due to an unexpected increased rate of cancer relapse (33.5% with azithromycin vs 22.3% with placebo; *p* = 0.002) and death (2-year survival of 56.6% vs 70.1% in the placebo group; *p* = 0.02) [106]. A subsequent study, where azithromycin was given for treating an established bronchiolitis obliterans syndrome many months to years after the allogeneic-HSCT, did not show increased relapse of the original malignancy but an increased number of secondary neoplasms; however, the reason of it is unknown [38]. It was proven that long-term low dose of azithromycin may impair the immune system’s function in response to antigens, [108,109,110] and alterations of the gut microbiota within the first months after allogeneic-HSCT were associated with an increased incidence of haematological relapse [100]. Gomez J and Duenas V, reported an interesting case of a CML patient that after an episode of infectious meningoencephalitis developed a rare blast crisis of the central nervous system, but whether the pathogens were responsible of the progression of the disease remains to be clarified [111].

Clearly, several questions are still unresolved, and our understanding remains superficial. Most critically, it is imperative that the mechanisms be understood in more detail, especially in the context of understanding what microbial traits drive an immunosuppressed phenotype incapable of detecting persistent leukaemic cells. Certainly, some microbes induce immunosuppressive responses in their host, creating a microenvironment favourable to their proliferation. For example, during lung infections, invading bacteria hijack the immune system by increasing immunosuppressive cell populations, including Tregs and MDSCs, and reducing the production of pro-inflammatory cytokines [112]. However, these have not yet been identified in the context of cancer relapse, let alone an ALL/CML relapse, and remain at odds with the predominantly “immunostimulatory” changes caused by cancer treatment which tend to drive exaggerated immune responses. The key to understanding this will be prospective, longitudinal studies that capture dynamic changes in the microbiota and immune system throughout and after treatment, paired with robust clinical and personal data need to account for the numerous confounding variables that affect the microbiota. If robust microbial signals are identified, this would warrant interventional strategies targeting the microbiota to decrease the risk of relapse.

## 6. Concluding Remarks and Future Perspectives

The gut microbiota has been studied in enormous detail in the past decades, highlighting its role during carcinogenesis and treatment toxicity. The role of the immune-mediated control in the progression of both CML and ALL was recently identified, but a direct link between dysbiosis and the dysregulation of the immune system in these diseases is missing. Understanding this connection can open the avenue to new therapies aimed to simultaneously reduce both cancer progression and chronic toxicities. Modulation of the cancer microbiota is an exciting field of research and potentially very feasible given the ease at which the microbiota can be modulated. Dietary intervention is able to induce significant changes in the gut microbiota composition in 24–48 h but can be difficult to implement in people with cancer due to difficulties eating due to oral mucositis, taste changes, and anorexia [113]. Pre- and probiotics may offer a more feasible method of intervention, but it remains unclear if these strategies are sufficiently powerful to support the microbiota after such profound damage. While there are some guidelines recommending the use of probiotics in specific patients, for example those by the Multinational Association for Supportive Care in Cancer, [114] probiotics do not have sufficient evidence to support their broad use [115]. In contrast, faecal microbiome transplantation (FMT) is an attractive strategy for restoring the gut microbiota composition after cancer therapy. It involves the transplant of faecal material from a healthy donor to a patient. In the context of cancer therapy, there is the unique opportunity to collect and bank baseline (pre-treatment) stool before starting therapy. This can then be used for autologous FMT, increasing the rate of colonisation, and decreasing the risk of disease transmission from third-party donors. While emerging in its experimental indications, FMT is currently only approved for *Clostiridum difficile* infection and more work is needed to refine the methodological considerations for use in cancer care [116]. Additionally, there is no general consensus about what a healthy microbiota is, whether there are risks of reintroducing a pro-carcinogenic microbiota via autologous FMT and what attributes are important to consider in donor FMT. As such, there are alternative methods of intervention including engineered FMT or microbial metabolites called postbiotics. Postbiotics are bacterial bioactive compounds, such as SCFAs, flavonoids, and taurine that can be delivered directly to the patient [117]. Screening of the gut microbiota metabolites is of current interest, in association with the sequencing of the whole microbiome and bacterial transcriptome. This will allow the identification of specific compounds associated to a specific microbiome and may facilitate the identification of postbiotics to be used in immune-mediated diseases. An increase understanding of the complex interaction between microbiota, immunity, and cancer relapse, is therefore required for the development of these new microbiome-based therapies. A successful clinical translation of these new finding requires a standardization of the microbiome-based methods for being used as gold standard practice in the laboratories worldwide.

## Figures and Tables

**Figure 1 microorganisms-10-00713-f001:**
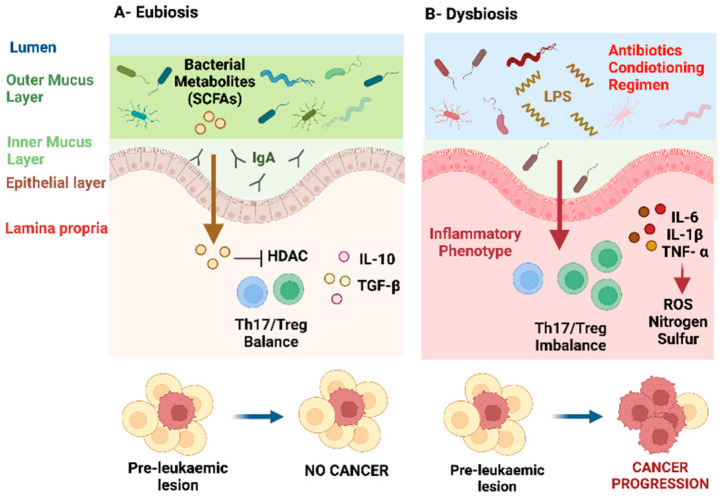
Involvement of the gut microbiota on leukaemia progression. (**A**) Eubiosis. Pre-leukaemic clones can arise in genetically predisposed individuals (for example *Pax5* heterozygosity and the *ETV6-RUNX1* fusion in acute lymphoblastic leukaemia, ALL), but an intact microbiota can protect from the development of leukaemia. The gut microbiota can modulate the host metabolism, inflammation, and immune responses, and it is involved in the maintenance of the intestinal barrier. Here is represented the distal colon containing two mucus layers: the outer mucous layer containing the gut microbiota, and a stratified adherent inner mucus layer essentially sterile. The microbiota produces metabolites, including Short Chain Fatty Acids (SCFAs). They are involved in the maintenance of the intestinal epithelial barrier, suppress inflammatory cytokines, and control host immunity through epigenetic changes. SCFAs can inhibit the Histone Deacetylase Activity (HDAC) inducing the differentiation of Forkhead box protein P3 (FOXP3)+ Regulatory T cells (Tregs), and maintaining a balance between Tregs and T-helper 17 (Th17) cells. Tregs produce the anti-inflammatory cytokines Interleukin (IL)-10 and Transforming Growth Factor beta (TGF-β). (**B**) Dysbiosis. Alterations of this delicate balance can induce disease progression. Antibiotics, a conditioning regimen, as well as a change in diet or medication, can induce a loss of beneficial bacteria, with increase in pathological bacteria. Lipopolysaccharide (LPS)-producing bacteria are increased, inducing mucositis, disruption of the mucus layer, and break of the epithelial barrier. Pathogens can therefore invade the lamina propria and activate immune responses, with imbalance of Treg and Th17 cells, and secretion of proinflammatory cytokines, such as IL-1β, IL-6, TNF-α. They in turn stimulate the production of reactive oxygen species (ROS), nitrogen and sulphur, causing oxidative damage, and leading to cancer progression. Created with BioRender.com. Licence obtained on 14 March 2022.

**Figure 2 microorganisms-10-00713-f002:**
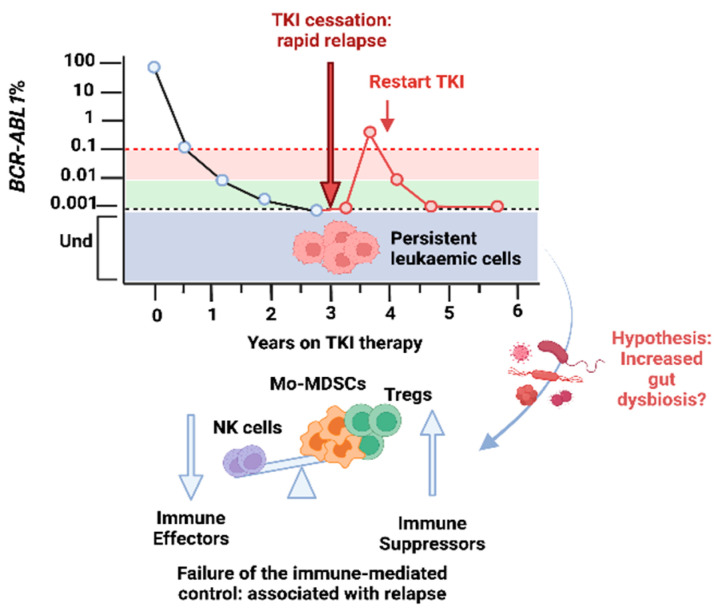
Molecular relapse in Chronic Myeloid Leukaemia (CML) after cessation of the Tyrosine Kinase Inhibitor (TKI) therapy. The response to TKI therapy is assessed by measuring the levels of *BCR-ABL1* transcript in the blood by a conventional real-time quantitative reverse transcription PCR and through the achievement of molecular milestones over the time of therapy. Major molecular response (in red) is defined by *BCR-ABL1* ≤ 0.1% and deep molecular response (in green) is defined by *BCR-ABL1* ≤ 0.01% or lower. Samples with *BCR-ABL1* mRNA levels less than 0.001% are considered undetectable (in grey), and this is the limit of detection of the method. With prolonged TKI therapy some patients have no longer detectable *BCR-ABL1* mRNA. Patients under TKI therapy for at least 3 years and in deep molecular response (green) for at least 2 years can cease their TKI to attempt treatment-free remission (black line). About 50% of patients who cease TKI therapy have a rapid molecular relapse, defined as loss of major molecular response (red line) and need to resume their TKI. More than 90% of them achieve again deep molecular response (green). Disease recurrence may be explained by the persistence of residual leukaemia cells not eradicated by the therapy, which provide a source of relapse and a bottleneck to cure. Growing evidence indicates the failure of the immunological surveillance of the residual CML cells associated with relapse. Gut dysbiosis is hypothesized to be involved in the creation of the immunosuppressive microenvironment which sustains cancer progression and relapse. NK cells: natural killer cells; T regs: FOXP3+ Regulatory T cells; mo-MDSCs: monocytic myeloid-derived suppressor cells. Created with BioRender.com. Licence obtained on 14 March 2022.

## Data Availability

Not applicable.

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
