# Peer review of "A Gut Instinct on Leukaemia: A New Mechanistic Hypothesis for Microbiota-Immune Crosstalk in Disease Progression and Relapse"

_microorganisms, 2022, doi:10.3390/microorganisms10040713_

Round 1

Reviewer 1 Report

The review article "A new gut instinct on leukaemia relapse" is a very comprehensive article with an extensive introduction on the treated diseases CML and ALL and an overview of the current state of knowledge on gut microbiota, carcinogenesis and relapse. What is impressive about this manuscript is the number of citations on the topic covered (121), but this is put into perspective as the first 51 references were only needed for the introduction. This is also reflected in the manuscript as a whole. Thus, the first half of the whole text does not deal with the influence of the microorganisms on the diseases but deals mostly with details of ALL and CML. For me as a reader, who had expected an overview of the gut microbiome, the text was clearly too long until reaching the actual topic. Here I would suggest a significantly shorter introduction, where certainly some details can be omitted. For example, the immunological changes during the progression of the diseases are of little interest here unless they are better linked to the influences of the microbiome. In this way, space can also be created to write point 4, which is about the actual topic, in a much more convincing and detailed way. In particular, the comment at the end of paragraph 4.2 "Beyond this, there has been limited investigation of the CML microbiota" implies that a review on this topic of the microbiome in CML may not be necessary at all. 

Minor comments:
Line 81 "...only occurring in 5-10 of patients..." Here you must mean 5-10 % of patients. 
Line 87: The text in parentheses is incomprehensible to a non-experienced reader. What is meant by BCR-ABL1 < 0.01%? Does this mean minimal residual disease? The limit here is 0.01% of what? These details are unimportant to the reviewed topic. The same concerns line 91.

Author Response

Dear Prof. Dr. Martin von Bergen,

Thank you for the opportunity to submit an amended manuscript. We thank both reviewers for their careful review and for their positive remarks on the scientific value of our work. Our responses to specific comments follow.

We trust that these changes have improved the clarity of the paper.

Yours sincerely,

Ilaria S Pagani, Govinda Poudel, Hannah R Wardill.

Reviewer 1

The review article "A new gut instinct on leukaemia relapse" is a very comprehensive article with an extensive introduction on the treated diseases CML and ALL and an overview of the current state of knowledge on gut microbiota, carcinogenesis and relapse. What is impressive about this manuscript is the number of citations on the topic covered (121), but this is put into perspective as the first 51 references were only needed for the introduction. This is also reflected in the manuscript as a whole. Thus, the first half of the whole text does not deal with the influence of the microorganisms on the diseases but deals mostly with details of ALL and CML. For me as a reader, who had expected an overview of the gut microbiome, the text was clearly too long until reaching the actual topic. Here I would suggest a significantly shorter introduction, where certainly some details can be omitted. For example, the immunological changes during the progression of the diseases are of little interest here unless they are better linked to the influences of the microbiome. In this way, space can also be created to write point 4, which is about the actual topic, in a much more convincing and detailed way. In particular, the comment at the end of paragraph 4.2 "Beyond this, there has been limited investigation of the CML microbiota" implies that a review on this topic of the microbiome in CML may not be necessary at all. 

We thank the reviewer for these valuable suggestions. We extensively edited the paper, hoping that these changes improved its clarity and value. All the changes are detailed in the "Tracked version of the manuscript". Here briefly our changes:

1- We followed the reviewer's advice shortening the introduction and condensing paragraphs 2 and 3 in the introduction.

2- We inserted paragraph 2 about "gut microbiota and regulation of the host immune system in homeostasis", giving an overview of the gut microbiome, with link to the immune system.

3- Paragraph 3 is now about "Immune-mediated mechanisms that drive relapse in CML and ALL". We condensed the immunological changes during the progression of both CML and ALL in one unique paragraph, linking them to dysbiosis and failure of the immune surveillance.

4- Paragraph 4 is now "Gut microbiota and carcinogenesis", where we described the association of dysbiosis with carcinogenesis.

5- "Microbial dysbiosis in cancer progression and relapse" We inserted the paragraph 5.1 "Changes in the gut microbiota during treatment and association with treatment toxicity". We performed new literature search. We inserted information about lipopolysaccharide and CML progression.

6- Paragraph 5.2 is "Microbial dysbiosis and relapse". We inserted new literature search about the role of lipopolysaccharide in cancer relapse. We further investigated how SCFAs modulate the immune system and reduce proliferation in a model of BCR-ABL-positive cell lines. We additionally found an interesting report describing the insurgence of a rare blast crisis in the nervous system in a CML patient after an episode of infectious meningoencephalitis.

7- We edited figures 1 (Involvement of the gut microbiota on leukaemia progression) and 2 (Molecular relapse in Chronic Myeloid Leukaemia (CML) after cessation of the Tyrosine Kinase Inhibitor (TKI) therapy), including SCFAs and LPS in the figure, and incorporating a research question linking gut dysbiosis, to immunosuppressive microenvironment and CML relapse.

We hope that this edited "Hypothesis generating review" will now convince the readers about the crosstalk between the microbiota and the immune system in disease progression and relapse.

Minor comments:

These parts have been eliminated, because irrelevant to our story, thank you for the suggestions.

Reviewer 2

In this review the authors analyse the influence gut microbiota have in development and progression of both chronic myeloid and acute lymphoblastic leukaemia. They also describe how microbiota composition is altered after different treatments. This review contributes to better understanding of the recently established relationship between gut microbiota and cancer, more specifically haematological malignancies.

Comments:

  • It would be interesting to include the references that support the immunomodulatory capacity of microbiota y disease progression and relapse (line 56).

We thank the reviewer for these valuable suggestion. We extensively edited the paper, hoping that these changes improved its clarity and value. All the changes are detailed in the "Tracked version of the manuscript” and summarised to Reviewer 1.

References have been included as suggested.

" These changes have been associated with altered immune responses, emphasising the immunomodulatory capacity of the microbiota, which have already been implicated in disease progression and relapse.(19-21, 23)" Lines 76-78.

  • The title of the manuscript is quite pretentious, as the review applies for the progression of the disease and the response to treatment and only one section is focused in leukaemic relapses. In fact, the authors claim that “The idea that the microbiota may influence relapse is gaining theoretical traction, but certainly remains speculative”.

We changed the title to "A gut instinct on leukaemia: a new mechanistic hypothesis for microbiota-immune crosstalk in disease progression and relapse".

We changed the sentence “The idea that the microbiota may influence relapse is gaining theoretical traction, but certainly remains speculative” to "The idea that the microbiota may influence cancer relapse is gaining theoretical and experimental traction", inserting new evidence about the role of the microbiota and microbial metabolites on cancer relapse and modulation of the immune systems. This new information has been added at the lines: 348-257, 304-313, and 329-332.

We hope that this edited "Hypothesis generating review" will now convince the readers about the crosstalk between the microbiota and the immune system in disease progression and relapse.

Reviewer 2 Report

In this review the authors analyse the influence gut microbiota have in development and progression of both chronic myeloid and acute lymphoblastic leukaemia. They also describe how microbiota composition is altered after different treatments. This review contributes to better understanding of the recently established relationship between gut microbiota and cancer, more specifically haematological malignancies.

Comments:

  • It would be interesting to include the references that support the immunomodulatory capacity of microbiota y disease progression and relapse (line 56).
  • The title of the manuscript is quite pretentious, as the review applies for the progression of the disease and the response to treatment and only one section is focused in leukaemic relapses. In fact, the authors claim that “The idea that the microbiota may influence relapse is gaining theoretical traction, but certainly remains speculative”.

Author Response

(The authors gave the same response as above.)

Round 2

Reviewer 1 Report

The present manuscript now shows much improvement and gets to the point addressed in the title. All concerns have been adequately addressed. Therefore, I rate this manuscript as eligible for publication in Microorganisms. 

Since this review is titled in the abstract as "Hypothesis-generating review", I would like to suggest to explicitly state the generated hypotheses again in a bullet point manner at the end.